# Changes of DNA Damage Effect of T-2 or Deoxynivalenol Toxins during Three Weeks Exposure in Common Carp (*Cyprinus carpio* L.) Revealed by LORD-Q PCR

**DOI:** 10.3390/toxins13080576

**Published:** 2021-08-19

**Authors:** Rubina Tünde Szabó, Mária Kovács-Weber, Krisztián Milán Balogh, Miklós Mézes, Balázs Kovács

**Affiliations:** 1Institute of Animal Husbandry, Gödöllő Campus, Hungarian University of Agriculture and Life Sciences, H-2100 Gödöllő, Hungary; szabo.rubina.tunde@uni-mate.hu; 2Department of Feed Toxicology, Institute of Physiology and Nutrition, Gödöllő Campus, Hungarian University of Agriculture and Life Sciences, H-2100 Gödöllő, Hungary; balogh.krisztian.milan@uni-mate.hu (K.M.B.); mezes.miklos@uni-mate.hu (M.M.); 3MTA-KE-SZIE Mycotoxins in the Food Chain Research Group, Department of Physiology and Animal Health, Institute of Physiology and Nutrition, Kaposvár Campus, Hungarian University of Agriculture and Life Sciences, H-7400 Kaposvár, Hungary; 4Department of Molecular Ecology, Institute of Aquaculture and Environmental Safety, Hungarian University of Agriculture and Life Sciences, H-2100 Gödöllő, Hungary; kovacs.balazs@uni-mate.hu

**Keywords:** LORD-Q PCR, DNA damage, trichothecene mycotoxin, *Common carp*

## Abstract

The present study aimed to adapt a Long-run Real-time DNA Damage Quantification (LORD-Q) qPCR-based method for the analysis of the mitochondrial genome of *Common carp* (*Cyprinus carpio* L.) and detect the DNA damaging effect of T-2 (4.11 mg kg^−1^) and deoxynivalenol (5.96 mg kg^−1^) mycotoxins in a 3-week feeding period. One-year-old *Common carp* were treated in groups (control, T-2 and DON). The mycotoxins were sprayed over the complete pelleted feed, and samples were taken weekly. Following the adaptation of LORD-Q PCR method for the *Common carp* species, the number of lesions were calculated to determine the amount of DNA damage. In the first and second weeks, the T-2 and the DON treated groups differed significantly from each other; however these differences disappeared in the third week. There was a significant difference in the DNA lesion values between weeks 1 and 3 in the deoxynivalenol-contaminated groups. While in the T-2 treated groups, the DNA lesion values were significantly reduced on weeks 2 and 3 compared to week 1. The results suggested that the trichothecene mycotoxins have a relevant DNA damaging effect.

## 1. Introduction

Food and feed can be contaminated with different types and amounts of mycotoxins, and they have various damaging effects [1]. T-2 toxin belongs to the group of type A trichothecenes that are produced by the phytopathogenic *Fusarium moulds* (i.e., *Fusarium sporotrichioides*, *F. poae*). It is known that T-2 toxin is the most toxic among the type A toxins (in vivo and in vitro, too) because of the trichothecene structure (epoxy sesquiterpenoid moiety) [2,3]. Deoxynivalenol (DON), as a member of the type B trichothecene group, is the most prevalent mycotoxin in cereal grains worldwide [2,3,4,5].

Cereals are increasingly used for fish feed production because of the growing in-tensification of aquaculture, the rising utilization of artificial feed and the replacement of fishmeal; therefore the risk of mycotoxin contamination has increased [6,7]. *Common carp* (*Cyprinus carpio* L.) is one of the most important omnivorous species in freshwater aquaculture, although limited information is available about the effect of T-2 or DON mycotoxins in this species, especially about the DNA damaging effect [6,8]. Long-term treatment with 2.45 mg T-2 kg^−1^ induced the activity of glutathione redox system in *Common carp*. The amount of glutathione (GSH) and glutathione peroxidase (GSHPx) were significantly increased on the 7th, 21st and 28th days. The growth rate was reduced but the concentration of the malondialdehyde (MDA) did not change [9]. During a four-week trial, Matejova et al. [6] did not observe mRNA expression changes on stress, detoxification or DNA damage influenced CYP2F2, CYP450, catalase, and *HSP60* and *HSP70* genes in carp. Although the lipid peroxidation was enhanced, the activity of glutathione S-transferase (GST) was significantly increased by 5.3 mg toxin kg^−1^. Pelyhe et al. [10] showed that a 4.11 mg T-2 kg^−1^ dose has an effect on the expression of *gpx4* genes in *Common carp*. *gpx4b* gene expression was upregulated on days 7, 14 and 21, while *gpx4a* was downregulated on days 14 and then upregulated on day 21. Considering DON, results of Pietsch et al. [11] showed that 352 μg DON kg^−1^ raised the catalase concentration and cytotoxicity in *Common carp*. DON caused damage and apoptosis in liver cells through lipid peroxidation and increased the MDA concentration in the kidney in a dose-dependent way [12]. Elevation of GSH amount was observed on days 7 and 28 at the concentration of 5.96 mg DON kg^−1^ [10]. The expression of the *gpx4b* gene was significantly upregulated on days 7, 14, 21 and 28. Comet assay analyses revealed that a significant increase in DNA damage (% tail DNA) occurred by aflatoxin B1 (4 mg/kg, 6 mg/kg) in *Common carp* [13]. 

The growing importance of genotoxicity testing and research of caused genetic dys-functions in different diseases requires new and trustful methods for the detection of amino acid injuries [14]. The most widely used methods can detect DNA damage in a global and sequence-independent way, such as the comet assay (single-cell gel electrophoresis), Halo assay or flow cytometry. However, these methods detect only one or a few types of DNA lesions [15]. If the nucleotide modifications disorder or inhibit the DNA polymerase, the DNA-synthesis-based methods can be used for the detection. The PCR-based methods represent more modern and adequate techniques for measuring DNA damage. LORD-Q PCR (Long-run Real-time DNA Damage Quantification PCR) is a highly efficent technique based on the principle that different DNA lesions halt the DNA replication and reduce the amount of PCR product [16]. It is a fast and quantitative analytical method that has many advantages. DNA damage can be measured on long DNA templates (3–4 kb) with a highly sensitive and sequence-specific manner (because this assay is primer-based) and allows the analysis of mitochondrial and nuclear genome [14,16] as well. LORD-Q PCR requires optimization to maximize the efficiency and the specificity of PCR, to get a robust, easy-to-control and high-throughput DNA damage quantification method [17].

Endogenous (i.e., normal cellular metabolism) and exogenous (i.e., genotoxic chemicals, mycotoxins, radiation) agents can cause different DNA modifications in living cells, such as oxidation products, dimers, and single or double-strand breaks [1]. The cell ap-plies numerous mechanisms to repair DNA damages so the integrity of nuclear and mitochondrial DNA can be maintained. If the damage recognition or the repair fails, the accumulation of DNA lesions occurs. The mitochondrial DNA (mtDNA) is susceptible to mutations, mainly because of the mitochondrial production of reactive oxygen species (ROS), which can cause oxidative lesions in the mtDNA. In the case of the mitochondria, the DNA repair capacity is lower than the capacity of the nucleus, and the copy number of mtDNA is much higher [16,18]. These make it more suitable for testing the DNA damage effects.

In the present study, we describe the changes in the DNA damaging effect of T-2 and DON mycotoxins in a 3-week experimental period in *Common carp* (*Cyprinus carpio* L.) For this purpose, a new LORD-Q PCR assay was optimized to detect and calculate the number of DNA lesions in mtDNA in vivo. In conclusion, LORD-Q PCR assay is a sensitive, suitable and robust technique to detect sequence-specific DNA damage. DON and T-2 mycotoxins can trigger DNA lesions in a three-week-long period.

## 2. Results

### 2.1. Validation of the LORD-Q PCR Assay in Common Carp Species

In order to adapt and develop the LORD-Q PCR quantification method for the common carp species, four precast qPCR mix and components such as DNA polymerases (e.g., RANGER DNA Polymerase, Bioline, London, England or DreamTaq, Fermentase (Waltham, MA, USA)) with different DNA-intercalating dyes (e.g., EvaGreen, Biotium (Fremont, CA, USA), SYBR Green, Roche Molecular Biochemicals, Basel, Switzerland)) and additives (e.g., DMSO, ROX, Sigma, Burlington, MA, USA) were tested (data not shown). Finally, the 5× Hot FirePol^®^ EvaGreen^®^ qPCR Supermix (Solis Biodyne, Tartu, Estonia) reaction mixture was chosen based on test PCR reaction because the preheat incubation of the used Hot FirePol^®^ DNA polymerase inhibits the extension of non-specifically annealed primers and primer-dimers [19]. Seven long and five short primer pair candidates with different amplicon sizes (3401–6876; 88–242) and different template concentrations (0.5–50 ng) were tested according to PCR specificity and efficiencies. The optimal primer pairs were the F2-R4.F with F2-R4.R for short (156 bp amplicon) and F3-R2.F with F3-R2.R for long (3436 bp amplicon) (Table 3). The gel electrophoresis (Figure 1) and melt curve analysis (Figure 2) did not detect unspecific products.

### 2.2. Mortality and Disorder Results

During the three-week feeding period, elevated mortalities were detected in the DON or T-2 contaminated diets compared to the control group. A high mortality rate was observed in the T-2 group, while the DON group had a lower rate as shown in Table 1. Fish necropsy did not reveal any specific disorder or symptoms; however, all specimen showed moderate liver necrosis and intestine inflammation.

### 2.3. Lesion Rates by LORD-Q PCR Assay

The LORD-Q analysis revealed that the DNA damage is not constant (Table 2). In the case of both the T-2 and the DON mycotoxins, mitochondrial DNA damage decreased over time. The average lesion value of the first week was significantly different from the second (*p* = 0.019) and third (*p* = 0.017) weeks in the T-2 mycotoxin experiment, while the change was not significant between weeks 2 and 3 in DNA damage (*p* = 0.937).

The decreasing tendency was slightly different at DON. The average lesion value significantly differed between the first and third weeks (*p* = 0.025), but the decreases were not significant, neither between the first and second (*p* = 0.476) nor between the second and third weeks (*p* = 0.191) during the DON treatment. 

The T-2 and DON mycotoxin-treated groups significantly differed from each other and from the control group in week 1 (*p* = 0.026) and week 2 (*p* = 0.008); this difference was eliminated in week 3 between the mycotoxin groups (*p* = 0.767) and between the treated and the untreated groups.

## 3. Discussion

DNA damage is continuously generated by different endogenous and exogenous agents and their metabolites [20]. It is important to be aware of the level of DNA damage and their evolution over time to understand the molecular effects, the mechanism of action and key elements of the process in many agents to connect the knowledge about oxidative stress, DNA structure changes, repair mechanisms and carcinogenesis. 

Until now, many methods were developed to measure the level of DNA damages, as comet assay [13], repair assisted damage detection [21], Southern blot analysis [22], high-performance liquid chromatography (HPLC) [23], halo assay [24], and flow and imagecytometry [25]. Nevertheless, the sensitivity of the commonly used and conventional assays is low and measures the DNA damage in a global and sequence-independent way. Another problem is that the results of these methods are not comparable because different assays determine different parameters of DNA damage [15]. All these techniques require huge processing time and/or high amounts of genomic DNA samples. The development or adaptation of a method to quantify the locus-specific DNA damage can help to understand the molecular mechanisms caused by endogenous or exogenous agents. These problems are solved by LORD-Q PCR assay [17,26]. However, it is necessary to adapt this method for all species to produce comparable results.

We successfully adapted the LORD-Q PCR assay for measuring mtDNA damage caused by T-2 or DON mycotoxin in the *Common carp* species. The sensitivity of the presented LORD-Q PCR is similar to the semi-long run real-time PCR (SLR PCR) [18]. Zhu and Coffmann [18] reported that longer amplicons are more sensitive to low levels of DNA damage, and LORD-Q PCR uses a longer amplicon, in contrast to the SLR PCR. Moreover, Ref. [14] proved that LORD-Q PCR is more sensitive and effective than SLR PCR and comet assay. Even sublethal DNA damage lesion rate caused by concentrations of exogenous agents can be measured. DNA damages (lesions) of mtDNA are caused by a wide range of diseases, drugs and agents [14,27] because of the high copy number, the maternal inheritance and the role of mtDNA (e.g., redox homeostasis, energy metabolism), which is a common and frequently used target in toxicology and ecotoxicology. Considering all this and the assumption of lower or moderate DNA damage effects of the analyzed toxins doses, LORD-Q PCR is a well-founded choice [14].

The toxic effects of different mycotoxins are influenced by sex, age, species, dose and exposition time [28,29,30]. Aflatoxins are less toxic to brine shrimp larvae than trichothecenes. Although DON, nivalenol and 3-acetyl-DON are less toxic than HT-2 toxin, DAS and T-2 mycotoxin [31]. Many studies demonstrated that the acute toxicity of DON is lower than T-2 mycotoxin [29,32,33]. An opposite trend was observed in this long-term study because DON generated higher DNA lesion rates every week than T-2 mycotoxin in the hepatopancreas. The liver is responsible for the detoxification of mycotoxins (DON and T-2 mycotoxins); in this context, previous studies described the histological, morphological, or molecular changes in this organ [34]. Rainbow trout (Oncorhynchus mykiss) is one of the most sensitive species to the exposure of DON mycotoxin; on the other hand, *Common carp* is also a frequently used toxicology species to screen genotoxic effects by different methods (e.g., fish micronucleus test) [9,31]. 

Oxidative stress is one of the main reasons for DNA damage. The increased production of ROS (reactive oxygen species) leads to oxidatively induced DNA damage in cells [35,36]. Correlations were found between oxidative stress biomarkers (lipidperoxidation parameters such as MDA, glutathione peroxidase, superoxide dismutase and antioxidant capacity) and DNA damage level [37]. T-2 toxin treatment (0.1, 0.15, 0.2 and 0.3 µmol/L) significantly increased the amount of ROS in zebrafish embryos at 24 h [38]. Oxidative stress, as an effect of trichothecenes, was demonstrated during a long-term (4 weeks) feeding trial of T-2 and HT-2 toxins in *Common carps*. Ref. [9] observed the elevation of reactive oxygen metabolite concentration in the T-2 (2.45 mg kg^−1^ feed) + HT-2 (0.52 mg kg^−1^ feed) treated groups during the first and second weeks. GSH concentration and GSHPx activity decreased compared to the control group only at the second week. MDA concentration was significantly reduced at the third week. Therefore, the antioxidant system was able to eliminate the harmful effect of T-2 toxin in *Common carp* liver. These trends were confirmed by our study with higher mycotoxin concentration. The DNA lesion rate significantly decreased at the second and third weeks compared to the first week in the T-2 toxin-treated group. However, Ref. [6] represented a controversial finding by investigating oxidative stress indicator and detoxifying enzymes. In that study, after a four-week-long trial, catalase, GST and TBARS parameters were enhanced significantly; however, the used toxin concentration was higher (5.34 mg T-2 kg^−1^ feed) than in this experiment (4.11 mg T-2 kg^−1^ feed). 

This study found that the DNA lesion rate was significantly higher in the DON treated group in the first and second weeks compared to the T-2 mycotoxin-contaminated groups. MDA as a main oxidative stress marker can indicate the presence of stress, which could be one of the reasons for DNA damage. Ref. [12] demonstrated that MDA value was significantly increased by DON treatment at the concentration of 953 µg kg^−1^ after four weeks, but a lower concentration (352 µg kg^−1^, 619 µg kg^−1^) was not different from the control group. At a lower DON concentration (5.96 mg kg^−1^), the DNA lesion rate significantly declined in the third week in the present study. With the same toxin concentration as in this experiment, Ref. [10] did not find effects on the MDA and GST values during the four weeks; on the other hand, a significant amount of DNA lesions can arise, as measured in this long-term trial. In our study, the presented mycotoxins (T-2 mycotoxin 5-fold, DON 3-fold) caused high mortality, and Ref. [10] found the same trend. The mortality may be related to liver necrosis and the inflammatory effect of T-2 or DON, but these mycotoxins can inhibit the DNA and protein synthesis too. 

In conclusion, this study demonstrated a sensitive and suitable PCR method (LORD-Q PCR) for DNA damage quantification in *Common carp* hepatopancreas samples in vivo. This technique is a valuable, cost-effective and robust tool to detect sequence-specific DNA damage effects [14,16] and therefore allows valuable application in the field of toxicology and ecotoxicology. T-2 and DON mycotoxins triggered DNA lesions. However, DON induced significantly more lesions in the first and second weeks than T-2 toxin. Trichothecenes, especially DON, are a common contaminant of grain, feed and food [29], and little is known about the ecotoxicological impact and consequence of these mycotoxins; hence, further investigations are needed.

## 4. Materials and Methods

### 4.1. Mycotoxin Production and Analysis

T-2 toxin was produced by *Fusarium sporotrichioides* (NRRL 3299) and DON by *Fusarium graminearum* (NRRL 5883) strains on corn substrate by the methods of [39]. T-2 and HT-2 content of feed was measured based on the method of [40], and DON and 15-acetyl DON concentration was determined by HPLC after immunoaffinity cleanup according to the method of [41].

### 4.2. Animal and Experimental Design

One-year-old Szarvasi P34 hybrid *Common carps* (*Cyprinus carpio* L.) (n = 108) were obtained from a commercial fish farm (ÖKO 2000 Ltd., Akasztó, Hungary). After one week of acclimatization, carps were sorted randomly into control and two experimental groups (T-2 and DON) into six aquaria (150 L each). Each of six aquariums was used in a semi-static system with dechlorinated (water was aerated for at least 1 day before use, and then the aeration was continuous) tap water. The water change ratio was 50% on every second day. The light regimen was 12:12 h light/dark period, and the water temperature was 18 ± 1 °C. The dissolved oxygen concentration was 8 ± 1.1 mg L^−1^, the pH was between 7.2 and 7.5 and the average total ammonia level was 0.2 ± 0.03 mg L^−1^ during the experiment. These parameters were continuously inspected: temperature, daily with laboratory thermometer (±0.1 °C); pH and dissolved oxygen with HACH HQ30d Portable Multi-Parameter Meter and total ammonia level with HANNA HI 83203 Multiparameter Photometer. The bodyweight of carps was 23.26 ± 6.86 g at the start of the experiment. The animals were fed with extruded, slowly sinking growth feed for carp (GARANT Aqua Classic™, Garant-Tiernahrung GmbH, Pechlor, The Republic of Austria). The nutrient content (on dry matter basis) of the diet was 51.5% nitrogen-free extract, 30% crude protein, 7.5% crude ash, 7% crude fat and 5% crude fiber. Mycotoxin-containing fungal culture was dissolved in 50 mL ethanol for T-2 toxin and in 150 mL water for DON and sprayed over 3 kg of complete pelleted feed. The forage was kept at 50 °C overnight to eliminate the ethanol or the water content. The control diet was sham-treated with 25 mL ethanol and 75 mL water and prepared than the two mycotoxin treated diets. The control diet had <0.02 mg kg^−1^ T-2, <0.02 mg kg^−1^ HT-2, <0.02 mg kg^−1^ DON and <0.02 mg kg^−1^ 15-acetyl DON mycotoxin concentrations. The first experimental diet contained 4.11 mg T-2 kg^−1^ + 0.49 mg HT-2 kg^−1^ and the second 5.96 mg DON kg^−1^ + 0.33 mg 15-acetyl DON kg^−1^. The carps were fed two times a day (at 8 a.m. and 4 p.m.) with two equal portions. The feeding intensity was 1% of the bodyweight, which was determined weekly. The mortality was recorded every day at the same time. The feeding experiment was lasted for 3 weeks.

### 4.3. Sampling and DNS Isolation

Six randomly chosen carps were weighed and exterminated from the three groups weekly. All fish were over-anaesthetized with clove oil (*Syzygium aromaticum*) and decapitated before the sample collection. Hepatopancreas samples were taken into 1.5 mL collection tubes, frozen in liquid nitrogen immediately and stored at −80 °C until DNA isolation. Total DNA was purified using MagAttract^®^ HMW DNA Kit (Qiagen GmbH, Hilden, Germany) according to the manufacturer’s protocol. The DNA quantity and quality of the samples were verified by agarose gel electrophoresis and by NanoPhotometer (Implen GmbH, Munich, Germany) measurements. The isolated DNA showed a high purity (A_260_/A_280_ > 1.8) and was stored at −20 °C.

### 4.4. LORD-Q PCR Assay

The LORD-Q PCR for DNA damage quantification was carried out in a StepOnePlus™ Real Time PCR System (Thermo Fisher Scientific, San Jose, CA, USA) in low-profile, clear PCR tubes and caps (Axygen, New York, NY, USA). The PCR conditions for the different primers and fragments were optimized to get specific products, which were verified by melting curve analysis (StepOne™/StepOnePlus™ Software v 2.2) and agarose gel electrophoresis (Figure 1). The PCR amplification was monitored by real-time measurement of the intercalation of the fluorescent dye EvaGreen into the double-stranded DNA. The PCR reaction mix consisted of 5× Hot FirePol^®^ EvaGreen^®^ qPCR Supermix reaction mixture (Solis BioDyne, Tartu, Estonia), 3.3 pM/µL primers (specific for the short or the long fragment) and 10 ng of template DNA in a total volume of 10 µL per well. All primers for the PCR assay were designed by Primer3Plus [42] based on the reference sequence of *Cyprinus carpio* mithocondrial genome (Access no.: NC_001606). The primers were synthesized and purified by Integrated DNA Technologies (Leuven, Belgium). The sequences of the most optimal primers are shown in Table 3. The cycling conditions were as follows: a pre-incubation phase of 95 °C for 12 min was followed by 40 cycles of 10 s at 94 °C, 20 s at 59 °C, and 30 s at 72 °C (small fragment) or 4 min at 72 °C (large fragment). Each sample was assayed in quadruplicate. Five dilution points were used to obtain the PCR efficiency.

### 4.5. Data Analysis

Data analysis was based on the measurement of the cycle threshold (Ct) by the StepOne™/StepOnePlus™ Software v 2.2. The average Ct value from quadruplicate PCR was used. The difference between long and small fragments was calculated by ΔCt (control versus each treated condition). The DNA damage was calculated for lesion per 10 kb DNA by including the size of the long fragment as described by [17].
Lesion rate (Lesion per 10 kb DNA) = (1 – 2 − (Δlong − Δshort)) × (10,000 (bp))/(size of long fragment (bp))(1)

The statistical analysis was carried out using the R software package [43]. The normal distribution of the groups was checked by Shapiro–Wilk test. The different groups were compared with one-way ANOVA tests followed by Tukey post hoc test, with *p* ≤ 0.05 level of significance.

## Figures and Tables

**Figure 1 toxins-13-00576-f001:**
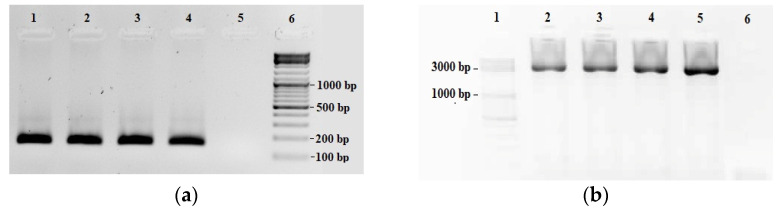
Specificity of the short (**a**) and long (**b**) fragments by agarose gel electrophoresis. Agarose gel (1.5%) electrophoresis of real-time PCR products with ethidium bromide staining; template DNS is 10 ng of template. (**a**) aAmplified products of the short fragment, the expected product size was 150 bp. Loading well 1–2 from T-2, loading well 3–4 from DON treated group (week 3). Loading well 5 is the negative control sample, loading well 6 is the size marker (GeneRuler^TM^ 100 bp DNA ladder). (**b**) Amplified products of the long fragment, the expected product size was 3400 bp. Loading well 1 is the size marker (GeneRuler ^TM^ 100 bp DNA ladder). Loading well 2–3 is from T-2, loading well 4–5 is from the DON treated group. Loading well 6 is the negative control sample.

**Figure 2 toxins-13-00576-f002:**
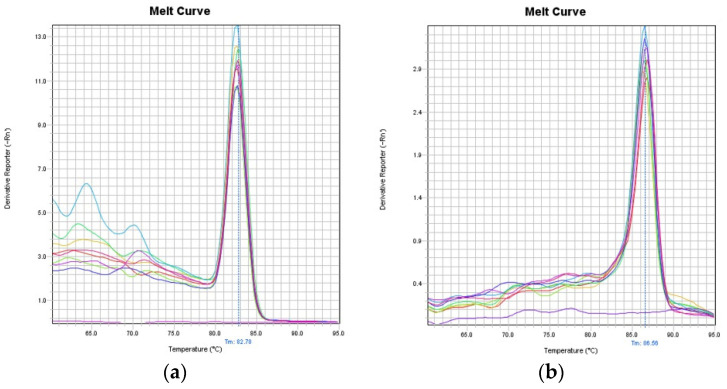
Amplification plot and melt curve of the short (**a**) and long (**b**) fragments by StepOne™/StepOnePlus™ Software (**a**) amplification plot, (**b**) melt curve of short fragment: two samples from T-2 group (2 blue line), two samples from the DON group (purple, pink line), one control sample (red, yellow, two green line) in the quadruplicate from week 3. The negative control is the magenta line. (**c**) Amplification plot of the long fragment: one control (yellow, two green, red) and one treated (two purple, two blue) samples in quadruplicate. (**d**) Melt curve of the long fragment: two samples from T-2 group (2 blue line), two samples from DON group (magenta, pink line), one control sample (red, yellow, two green line) in quadruplicate from week 3. The negative control is the purple line.

**Table 1 toxins-13-00576-t001:** Mortality-increasing effect of mycotoxins in the experimental groups during the trial (dead specimens/week).

	Control	DON	T-2
Week 1	0	1	1
Week 2	0	3	3
Week 3	2	4	4
Mortality rate %	5.5	16.7	27.8
Liver necrosis	0%	100%	100%
Intestine inflammation	0%	100%	100%

**Table 2 toxins-13-00576-t002:** The fold range of DNA lesions at groups treated with different mycotoxins compared to the control during the trial time by LORD-Q PCR.

Week	Control	T-2		DON	
	Mean	Mean	S.D.	Mean	S.D.
1.	0 ^A^	1.33 ^bB^	0.22	1.87 ^bC^	0.42
2.	0 ^A^	0.74 ^aB^	0.06	1.47 ^abC^	0.43
3.	0 ^A^	0.68 ^aA^	0.05	0.82 ^aA^	0.27

^a,b^: different superscript letters show significant differences (*p* ≤ 0.05) between weeks in T-2 or in DON groups by Tukey test. ^A,B,C^: different superscript letters show significant differences (*p* ≤ 0.05) between groups in the same week by Tukey test.

**Table 3 toxins-13-00576-t003:** Primers and their PCR parameter.

Primer Name	Sequence (5′ to 3′)	Amplicon Size (bp)	Position on the mt Genome (bp)	PCR Efficiency %	R^2^
F2-R4.R	AAGCACGGATCAGACGAACA	156	5887–6042	94	0.99
F2-R4.F	CACGCAGGAGCATCAGTAGA				
F3-R2.R	TAGGCTGGATAATAGGGTTGC	3436	647–4082	86	0.98
F3-R2.F	CCGTTCAACCTCACCACTTCT				

## Data Availability

The data presented in this study are available on request from the corresponding author (Mária Kovács-Weber).

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
