# Peer review of "Changes of DNA Damage Effect of T-2 or Deoxynivalenol Toxins during Three Weeks Exposure in Common Carp (Cyprinus carpio L.) Revealed by LORD-Q PCR"

_toxins, 2021, doi:10.3390/toxins13080576_

Round 1
Reviewer 1 Report
The paper deals with the effect of two toxins DON and T-2 on DNA damage from hepato-pancreatic samples of common carp. The introduction provides a good background. Some modifications to improve the paper are needed prior publication.
==> Introduction : line 26-29 : references 2/3/4/5 used to present DON and T-2 toxins are very specific to chicken or fish studies and do not reflect the toxicity or prevalence of those toxins in cereal grains. Some more general references such as McCormick a,d al, 2011 Trichotecenes : From single to complex mycotoxins or Bennett and Klich 2003 Mycotoxins should be used.
Line 39 : reference to paper 4 seems to be wrong as this study deals with the effect of DON mycotoxin and not T-2 toxin. Please add the right reference.
==> Results: line 93 suppliers of the TAQ should be added as already done for the dyes and additives
Figure 1 : Negative controls and explanations about the samples should be added (which sample concentrations…).
Figure 2 : Presented results are different for the short and the long fragment. In both cases it should be interesting to present one negative control, one treated sample and one control sample to fully explain the validation of the methodology.
Table 1 : The table shows numbers of dead animals whereas line 122-123 mortality rates are indicated. Authors should present in the table the Numbers of animals for each conditions, the Numbers of dead animals and the corresponding mortality rates for a better understanding.
==> discussion : line 172-173 reference dealing with mtDNA should be added
=> materials and methods : line 260 how were the tested concentrations chosen?
In table 3, for a better understanding, R² and number of serial dilutions points should be added to PCR efficiency
==> Globally : scientific names and genes shoud be in italic
A lot of references in the introduction and in the discussion explain the link betwween DNA damage and redox activity. Thus it would be very interesting to study catalase or GST activity as it seems to be really relevant to the study and will reinforce the discussion.
Reviewer 2 Report
Comments to manuscript #1323958 with the title „Different changes in the DNA damage effect of T-2 or deoxynivalenol toxins during three weeks exposure in common carp 3 (Cyprinus carpio L)”.
The authors describe the adaption of the long-run real-time DNA damage quantification (LORD-Q) method for the detection of DNA lesions to common carp. Using the highly sensitive LORD-Q method the authors analyze the effect of 3 week exposure to toxins (deoxynivalenol or T-2) on mtDNA of hepatopankreas samples from common carp. The authors conclude from their results that the toxins induce mtDNA damage in hepatopankreas mitochondrial DNA.
This reviewer recommends revision of the manuscript before acceptance.
General comment: it would be interesting to know whether nuclear DNA is damaged and to what extend.
The authors make the interesting observation that necropsy identifies inflammation. It would also be interesting to know whether inflammation is caused by necrotic tissue or pathogens and if so, which pathogens take advantage of the situation.
Specific points.
- Figure 1: marker size (bp) must be displayed. Samples need unique identification (throughout figures). A legend with sample ID for real-time PCR results must be included (also in figure 2).
- Table 1/2: A bar graph to visualize results is highly recommended.
- Line 86: insert “in” – lesions “in” mtDNA
- Line 102 and 289: “best” – please rephrase, e.g. “optimal”, “most efficient/specific”
- Line 104: “lack of non-specific” – please rephrase, e.g. “did not detect unspecific amplification”.
- Line 122: delete “was”
- Line 133 “a little bit” – please rephrase
- Line 172: delete "h" in: "mathernal" to make "maternal"
Minor points:
The authors are advised to check pdf-conversion of their manuscript. Throughout the text words are interrupted by hyphens.
Reviewer 3 Report
This manuscript presents a study of DNA damage by T-2 or DON to common carp using the LORD-Q PCR approach. The topic is of great interest to Journal readers. However. the writing quality of the entire manuscript needs to be improved for publication.
Some comments to improve the manuscript are:
- The title does not reflect the research work well. It could be something like “The DNA damage of T-2 or DON toxins to common carp revealed by LORD-Q PCR”?.
- The manuscript needs extensive editing to improve its English. For example:
Line 4, Cyprinus carpio L should be “ Cyprinus carpio L.”
Line 7 , 4.11 mg kg-1 should be “ 4.11 mg kg-1”
Line 12-13, … different from each other, please specify what is different from what.
Line 14, please specify lesion value. ( is the DNA lesion value?)
Line 31 replace-ment, please remove - from this word. There are many cases like this across the entire manuscript. Please correct them.
Line 57 widely used methods? Also, remove “are” from this sentence.
Line 86, Vivo should be italic.
Line 92: please specify how many precast qPCR mixes have been tested.
Line 274, (A260/A280>1.8) should be written as “A260nm/A280nm>1.8”
- In Fig. 1, the molecular size of DNA ladder should be labeled.
- In Fig. 2, what samples did the curves with different colors represent?
- Table 1, what did these numbers exactly represent?
- Line 141, do the lesions here refer to DNA lesions? Please specify it?
Round 2
Reviewer 3 Report
Some minor suggestions for further improvement:
- In Fig. 1, the molecular size of some bands in DNA ladder should be labeled. The loading well should be labeled by numbers.
- To make sure all revision was correctly made for example in Line 57 "usedable"?
Author Response
Dear Reviewer,
Thank you for the new suggestions. I resubmit to you the revised version of manuscript ‘Changes of DNA damage effect of T-2 or deoxynivalenol toxins during three weeks exposure in common carp (Cyprinus carpio L) revealed by LORD-Q PCR’.
I have responded to each suggestion below.
- In Fig. 1, the molecular size of some bands in DNA ladder should be labeled. The loading well should be labeled by numbers. – I inserted some relevant bands in DNS ladder (3000 and 1000 bp for long; 1000, 500, 200 and 100 bp for short fragments) and the numbers of the loading wells.
- To make sure all revision was correctly made for example in Line 57 "usedable"? – ’usable’ was corrected to used, in pdf and in word without correction, I see this sentence fine with used: The most widely used methods can detect DNA damage…
